# MarioGPT: Open-Ended Text2Level Generation through Large Language Models

**Shyam Sudhakaran**[1], **Miguel González-Duque**[*1], **Matthias Freiberger**[*1],
**Claire Glanois**[1], **Elias Najarro**[1], **Sebastian Risi**[1,2]
[1]IT University of Copenhagen, [2]modl.ai, Copenhagen
shyamsnair@protonmail.com, sebr@itu.dk

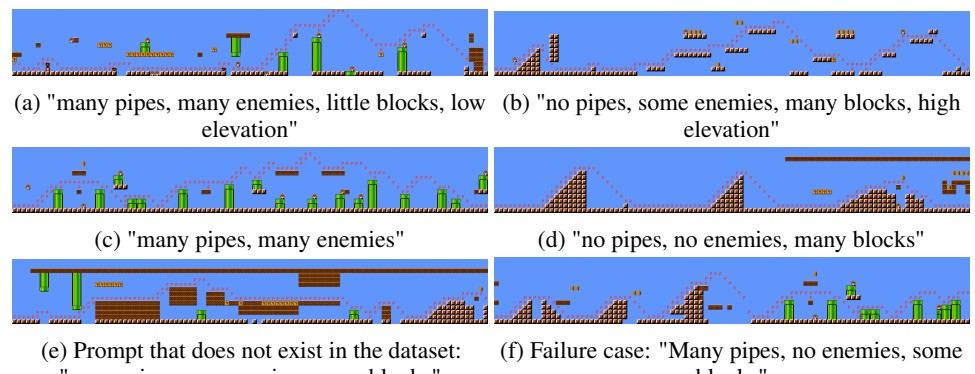

(a) "many pipes, many enemies, little blocks, low elevation"

(b) "no pipes, some enemies, many blocks, high elevation"

(c) "many pipes, many enemies"

(d) "no pipes, no enemies, many blocks"

(e) Prompt that does not exist in the dataset: "many pipes, no enemies, many blocks"

(f) Failure case: "Many pipes, no enemies, some blocks"

Figure 1: MarioGPT is able to successfully generate levels that follow the text prompt (**a–e**). Failure cases rarely happen: for example in (**f**) the model manages to generate many pipes and some blocks, but it still generates enemies even though it was prompted with "no enemies".

## Abstract

Procedural Content Generation (PCG) is a technique to generate complex and diverse environments in an automated way. However, while generating content with PCG methods is often straightforward, generating meaningful content that reflects specific intentions and constraints remains challenging. Furthermore, many PCG algorithms lack the ability to generate content in an open-ended manner. Recently, Large Language Models (LLMs) have shown to be incredibly effective in many diverse domains. These trained LLMs can be fine-tuned, re-using information and accelerating training for new tasks. Here, we introduce MarioGPT, a fine-tuned GPT2 model trained to generate tile-based game levels, in our case Super Mario Bros levels. MarioGPT can not only generate diverse levels, but can be text-prompted for controllable level generation, addressing one of the key challenges of current PCG techniques. As far as we know, MarioGPT is the first text-to-level model and combined with novelty search it enables the generation of diverse levels with varying play-style dynamics (i.e. player paths) and the open-ended discovery of an increasingly diverse range of content. Code available at https://github.com/shyamsn97/mario-gpt.

## 1 Introduction

Procedural Content Generation (PCG) refers to techniques that can automatically create game content, such as levels, maps, or characters [36]. Some of the benefits of PCG are an increase in the replayability of a game and reduced production costs.

---

*These authors contributed equally to this work.

37th Conference on Neural Information Processing Systems (NeurIPS 2023).

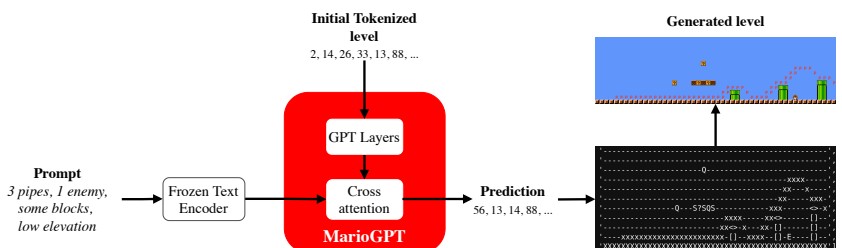

Figure 2: **MarioGPT prediction pipeline**. Our MarioGPT model is a finetuned version of the distilled GPT2 language model. Like GPT2, MarioGPT is trained to predict next token sequences. Levels are represented as strings, which are tokenized by a Byte-Pair Encoding, similar to the original GPT2 model. The level is split by columns and flattened into a single vector (or batch of vectors for multiple levels). To incorporate prompt information, we utilize a frozen text encoder in the form of a pretrained bidirectional LLM (BART), and output the average hidden states of the model's forward pass. This average hidden state is then used in the cross attention layers of the GPT2 architecture in combination with the actual level sequence being passed into the model.

Recently, developments in PCG and machine learning have started to influence each other in different ways [30]. PCG researchers are now incorporating machine learning-based approaches into their systems and models such as Generative Adversarial Networks (GANs) [13] can be trained to generate levels for games as diverse as Doom [10] or Super Mario Bros, training on levels from the Video Game Level Corpus [45]. However, current approaches in this field of Procedural Content Generation via Machine Learning (PCGML) [39] often rely on costly searching inside of the latent space of the underlying neural networks. It would be more desirable to being able to directly condition a generator to create levels with certain properties, ideally in natural language.

To address these challenges, we propose MarioGPT (Figure 2), a fine-tuned GPT-2 model trained to generate Mario levels. Our model demonstrates how LLMs can be combined with PCG techniques, enabling the effective creation of new and diverse levels through natural language prompts (Figure 1). Large language models (LLMs) trained on a diverse corpus such as the GPT-n family model [29], capture the statistical correlations of the human experience in the form of language correlations. Through this process, GPT acquires *knowledge* of how to represent and predict intricate sequences. We utilize this knowledge to provide our model with the ability to generate levels that incorporate simple artefacts as well as more complex relational properties. Surprisingly, a high percentage (88%) of MarioGPT generated levels are in fact playable.

Furthermore, we combine MarioGPT with novelty search [22], a diversity-seeking algorithm, to continually generate diverse levels in an open-ended manner. The combination of LLMs with algorithms such as novelty search opens up many interesting new directions for future research. We hope our work opens the door to more flexible and controllable PCG methods that can generate infinite content that is complex, diverse, and functional. To facilitate this, the code to run the experiments in this paper is publicly available at: `https://github.com/shyamsn97/mario-gpt`.

## 2 Background and Related Work

**Procedural Content Generation.** Procedural Content Generation (PCG) algorithms [36] deal with the automatic creation of game content (e.g. for level design, character generation, environment modeling, etc.). As reviewed in [36, 46], earlier works often focused on evolutionary computation [4], solver-based methods [37] or constructive generation methods (such as cellular automata, grammar-based methods, etc). More recently, deep learning for PCG [27, 39] has emerged as a promising approach to learning to generate high-quality game content in a data-driven manner, which is not only aesthetically pleasing but also functional and challenging. However, the diversity, originality and playability of the generated content in addition to the controllability of its generation, remain major challenges [39]. Our work aims to show how conditioned language models, paired with novelty-driven approaches to content generation [26, 2], could help tackle these shortcomings.

**Neural Network-based Level Generation.** Recent works in the space of video game level generation, particularly for Super Mario [2, 8, 45, 33, 35, 34], also leveraged neural network architectures to create levels. Beukman et al. [2] evolved neural networks in order to generate levels, while others [8, 45, 12, 34] performed evolution / search in the latent space of a trained generative model. These

works showed that guided sampling of the latent space of the learned generative model could result in a diverse set of levels.

Previous works that utilize a trained generative model [8, 45, 34, 12] also explored the abilities to control characteristics in generated levels. However, to do so these methods relied on searching the latent space (e.g. through quality diversity algorithms [28, 9] or evolutionary strategies [14]) for levels with specific target characteristics (e.g. a level with many pipes). This is a significant limitation because even though the generative models may represent a rich set of content, one has to search through its latent space to try to find the content that actually satisfies specific characteristics. MarioGPT is able to improve upon this limitation by incorporating text prompts into the actual generative process, allowing for easily controllable level generation. In other words, instead of searching for a level with specific characteristics, MarioGPT allows us to just ask for it. Concurrently to our work, Todd et al. [42] showed that LLMs can also be used to generate levels for other games such as Sokoban but their model did not allow for any text-prompting.

**Open-Endedness and Genetic Algorithms.** The open-endedness paradigm focuses on algorithms that can produce infinite innovation [24]. These open-ended algorithms are popular in the field of PCG, where designers and players both can benefit from diverse and never-ending content. However, PCG must balance the hard task of generating content with diversity as well as playability. Genetic algorithms (GA), a family of optimization algorithms that are inspired by the principles of natural selection, are commonly used as the backbone for more open-ended search methods. Because GAs allow the integration of multiple objectives, they are particularly suitable for achieving a balance between fitness and diversity.

In that regard, novelty search approaches [23] aim at finding the most novel solutions at each generation, in comparison to what has been seen (i.e. an archive of previously discovered highly-novel individuals). What makes novelty-search powerful, and motivated its use in this paper, is that it guides the generation towards increasingly diverse solutions in an open-ended fashion. Novelty search keeps track of solutions in an archive and measures diversity by the distance between their behavior characteristics (BCs) compared to that of their $k$ closest neighbors. This makes novelty search very flexible, allowing for the use of many different behavior characteristic types.

**Sequence Modelling and Transformers.** Classic approaches to sequence modelling using recurrent neural networks (RNNs) [31] and Long Short Term Memory (LSTM) networks [15] have traditionally been constrained by the fading memory of the network's state vector, as well as limited scalability due to the temporal interdependency of the operations. Transformers [44] address both challenges by applying associative attention [1] to learned reprojections of the windowed input sequence, which is commonly referred to as *self-attention*. These architectural innovations have enabled Large Language Models (LLMs) to learn from massive datasets. Additionally, such models have also shown to be effective in accelerated learning of down-stream tasks. Fine-tuning LLMs [7] involves using pre-trained model weights as a weight initialization for new tasks.

One particularly relevant use of pretrained / fine-tuned LLMs comes from the method *Evolution through Large Models* (ELM), proposed in Lehman et al. [21]. ELM utilizes an LLM diff model [3], which is trained on code diffs obtained by Github data, giving the model the ability to modify a code snippet based on a particular commit message. This diff model is used as a "mutation operator", for a GA that evolves a population of programs. The wide generative capabilities of the LLM produce diverse mutations, resulting in novel individuals that vary increasingly over the course of the GA.

## 3 Open-Ended Level Generation through LLMs

Here we present our complete approach to open-ended level generation through LLMs, which is composed of two parts. First, we introduce our prompt-conditioned model MarioGPT (Figure 2) in Section 3.1, which generates levels –encoded as text– given a natural-language prompt. Second, we detail how MarioGPT can be used in a novelty-search evolutionary loop (Figure 3) in Section 3.2, allowing the approach to produce a continual stream of diverse levels.

**Level Representation.** Mario levels are represented similarly to previous works [45, 8, 35, 33, 34, 12], using the levels provided in the Video Game Level Corpus (VGLC) [40]. We utilize a relatively small set of path-annotated levels, taken from Super Mario Bros. and Super Mario Bros.: The Lost Levels (in total 37 levels). For more details on specific tiles present, see Section 6.1 in the Appendix. These

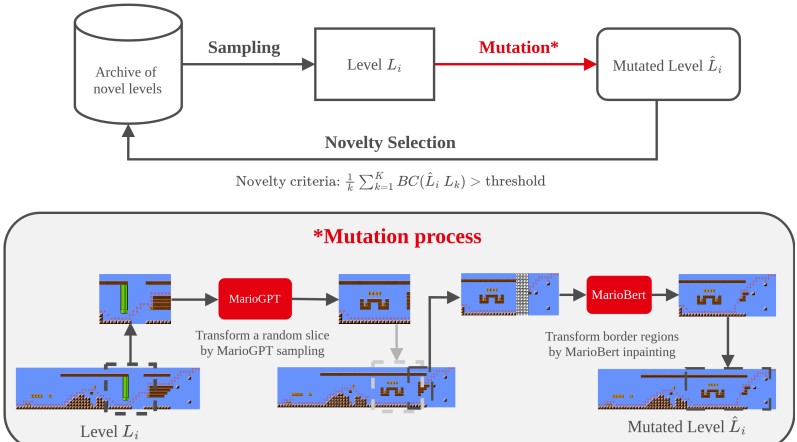

Figure 3: Novelty search setup and MarioGPT mutation operators. A level is sampled from a set of top elites in the archive, mutated, and, if novel enough, added to the archive. The mutation process involves two main steps: (1) Pick a random slice from the level and replace it with a new MarioGPT sample, using a random prompt. (2) Inpaint the border region with MarioBert to preserve path consistency.

levels are stitched together, to essentially make one giant level, allowing us to sample freely without worrying about the ends of the levels. Each tile is represented as a string. The string representation and characters are tokenized into discrete values using a Byte Pair Encoding tokenizer used in the original GPT2 model [29]. The tokenizer learns a mapping that maps each tile to its own unique token. One limitation from the dataset is the simplified representation of enemies. Even though levels contain many different enemies, each with different behaviors and features, the dataset represents them all as the same token.

## 3.1 MarioGPT Model

Our model, MarioGPT, is a prompt-conditioned unidirectional language model, optimized for long sequence level prediction. More precisely, MarioGPT's architecture relies on a distilled, lightweight version of GPT2 [29] transformer architecture called DistilGPT2 [32] [*]. Encoding slices of Mario levels as strings, similar to the approach taken in Summerville and Mateas [38], we can fine-tune this distilled version of GPT2 on predicting next tokens in Mario levels. To generate levels, we concatenate a window of previous 50 columns into a single vector and feed them into MarioGPT.

**Architecture**: MarioGPT's architecture is the same as the DistilGPT2 architecture, except the cross attention weights are utilized for prompting. Even though DistilGPT2 supports context lengths up to size 1024, we limit our context lengths to 700, as we found increasing it did little to increase performance. In total, MarioGPT has 96 million parameters (86 million of the original DistilGPT2 parameters and 10 million from cross attention weights). We train MarioGPT for 50,000 steps, sampling 4 random slices of levels at each iteration and optimize the model using the Adam optimizer [20]. In total, MarioGPT sees 200,000 training samples. Because the model is relatively small, it can be trained using a single Nvidia GeForce RTX 2080 Ti GPU.

**Prompting details**: In order to incorporate prompt information, we fine-tune the attention layers' cross attention weights, as illustrated in Figure 2. Prompts are encoded through BART [25], a frozen pre-trained language model. Prompts are passed through the frozen language model and the hidden states from the forward pass are averaged into a single vector. This average hidden state is then used in the cross attention layers of the GPT2 architecture in combination with the actual level sequence being passed into the model. We represent our prompts as combinations of specific features along with keywords that correspond to quantiles (e.g. none, little, some, many). This allows us to easily generate level/prompt pairs by counting corresponding tile values. For more details on the prompts, see Section 6.2 in the Appendix.

---

[*]We utilize the open source transformers library, specifically https://huggingface.co/distilgpt2

In addition, it is possible to use synonyms for words. For example, changing "many" to "a lot" or "a ton", produces similar results because the BART encoder can generalize well.

## 3.2 Open-Ended Mario Level Generation with Novelty Search

In the realm of PCG, it is important to not only generate levels with diverse physical features, but also levels that elicit a wide range of **player behavior**. When it comes to creating Mario levels, the focus is on the different paths a player can take to complete the level. This is often a challenge for many algorithms (such as [45, 8]) and requires the use of an external agent for evaluation. However, with MarioGPT, it is possible to generate diverse and controllable levels that approximate a realistic player path, reducing the need for an external agent and producing levels that are directly playable. To encourage diversity in generated levels, we integrate MarioGPT within a novelty search augmented genetic algorithm (NS-MarioGPT), where language-models play the role of mutation operators. As illustrated in Figure 3, NS-MarioGPT iteratively samples and mutates elite levels from an archive of generated levels.

**Novelty Search**: Mutated levels are only stored in the archive if they achieve a higher novelty score compared to the previous elites. The novelty score is measured as the mean distance between the behavioral characteristic vector of the levels and the behavioral characteristic vector of the $K$ closest elements from the archive ($K$-means). Our goal in level generation is to create paths that result in diverse player behavior, so we use **predicted** player paths as our basis for these behavior characteristics. More specifically, we are interested in the relative patterns of predicted paths. For instance, if a player character moves in a straight line on high elevated blocks, we want the path's representation to be close in behavior space to a path that moves straight in lower elevation. To achieve this, we represent the behavior characteristic as the normalized average of the predicted path's coordinates, allowing a smooth representation of paths (Figure 4). Thus the significance of a single block difference is reduced, making it harder for mutated levels to be added to the archive. This is desired because we don't want the archive to fill up with levels that only vary slightly from the existing levels in the archive. For all our novelty search experiments, we use a small neighborhood of size 4, which results in a behavioral characteristic of dimension 100. We initialize our archive with a small number of levels (30), as we found mutations are significant enough to generate a diverse set of levels without a big starting population.

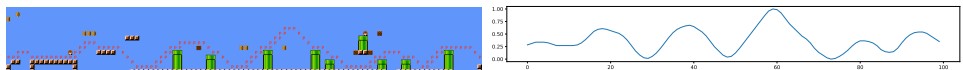

Figure 4: Novelty search behavior characteristic. Left: level, Right: smoothed moving average of generated path.

**Mutations:** The LLM-based mutation operation introduced in this paper (Figure 3) transforms a randomly picked slice of a level (a slice between $40 - 80$ columns) with a new MarioGPT prediction, guided by a random prompt. By itself, MarioGPT is able, through mutations, to produce a variety of levels with varying agent paths. However, because MarioGPT is a unidirectional model, we cannot guarantee that the new generated path is consistent with the rest of the level. To further improve path consistency, we incorporate a fine-tuned mask prediction model (which we call MarioBert), based on the Bert architecture. The BERT language model [7] is a bidirectional LLM that shows impressive performance in the task of mask prediction, which is analogous to image in-painting. This ability is ideal for our use case, where MarioBert is used to inpaint its border region after the newly sampled slice, smoothly joining the mutated slice and the rest of level. This can be observed in the second step of the "Mutation process" part of Figure 3.

## 4 Experiments and Results

### 4.1 Tile Prediction Accuracy

To measure how proficient MarioGPT is in generating levels and because the majority of tiles in these levels are air tiles, we focus on comparing non-air tile prediction accuracy. We compare to baselines: LSTM, as proposed in Summerville and Mateas [38] and MarioGPT that is trained

Table 1: Training Reconstruction Accuracy – Validation Set

| Model | Tile Acc. | Path Acc. | Promptable? |
|---|---|---|---|
| LSTM | 46% | 39% | NO |
| from-scratch-MarioGPT | 31% | 23% | YES |
| adapter-MarioGPT | 21% | 11% | YES |
| MarioGPT | 93% | 91% | YES |

from scratch (without using pretrained GPT2 weights), with results reported in Table 1. For all our baselines, we train for the same amount (200,000 samples). The results show that MarioGPT (using a pretrained GPT2 model) outperforms all other baselines with regards to tile prediction. In addition, training MarioGPT from scratch and training an adapter layer (a small multi layer network on top of the original prediction layer) results in models that performs worse than even the LSTM baseline (given the 200,000 training samples). These models were trained with minimal hyperparameter search, so their performance can likely be improved. However, as a tangential point, this shows a major benefit of fine-tuning pretrained models, which seem to require much less effort in regards to hyperparameters.

## 4.2 Measuring Playability of Levels

To test for playability, we deploy Robin Baumgarten's A* agent [43, 19] in 250 generated levels[*]. The reason for choosing Robin Baumgarten's A* agent for measuring playability comes from its performance on the 2009 Mario AI competition, where it beat handcrafted controllers and even simple evolved neural networks on getting the furthest in an infinite-level setting, as well as solving a corpus of levels [43]. We find that 88.4% of all MarioGPT-generated levels can be completed by the agent, and are therefore considered playable (compared to the best baseline, the LSTM, which achieves around 31% solvable levels). Moreover, we find that only one of the successful levels needed a retry with the A* agent. We further test whether the path generated by the model matches that of the A* agent to assess their feasibility. Table 2 shows the mean absolute error (MAE) between suggested and actual agent path for playable and not playable levels respectively. We see that for playable levels, the MAE between the path generated by the model and the actually taken path by the agent is $1.15$ tiles, i.e. paths are on average about 1 tile apart. For the non-playable levels, this average difference of taken paths is significantly higher with $4.56$ tiles. Thus, we can conclude that in playable levels, the agent mostly takes a similar path as the one generated by the model. The significantly higher MAE of $4.56$ in non-playable levels on the other hand indicates that the path generated by the models in these cases may not be feasible for the agent.

Table 2: Mean average error (MAE) between paths suggested by model and Baumgarten's A* agent. Results are averaged over 5 runs per level to account for minor stochastic variation in agent simulation. MAEs are computed between $y$ coordinates of path trajectories for every point on the $x$ axis (which goes across the level) both trajectories have visited.

| Playable | Not Playable | All |
|---|---|---|
| 1.15 | 4.56 | 1.56 |

Considering the MAE of $1.56$ tiles for paths in all levels, we can conclude that in the majority of the cases, the path generated by the model is similar to the path taken by an actual agent, and having the model generate a path through the level jointly with the level is an effective approach to obtain high-quality levels in terms of playability.

To investigate the quality of the generated paths further, we visualize the paths with the most, least and median overlap (i.e. the levels corresponding to the maximum, minimum and median values for the mean absolute error in height) as well as two interesting handpicked examples in Figure 5.

---

[*]Since the agent's performance depends on the available compute, we test for playability by running each level 5 times.

Figures 5d and 5e show that paths generated by MarioGPT tend to have more airtime than Baumgarten's agent in the sense that they only weakly take into account "gravity". This result may be attributed to the nature of the path annotations in the models training set. In Summerville et al. [40], the authors use an A* path solver to find a path through the level, while an actual agent, such as the one we used for comparison here, is more strongly bound by game physics (especially "gravity") and has to avoid enemies in the level. A second reason for non-playable levels can be seen in Figure 5c: Baumgarten's agent is spawned in a tight space from which it can not escape, while the model has generated a path that traverses beyond the actual level, again a path that would likely be suggested by a solver. We argue that these issues can in part be attributed to the paths in the training data stemming from a solver rather than an actual agent, and could be alleviated in future work by annotating the training data with the trajectories of actual agents.

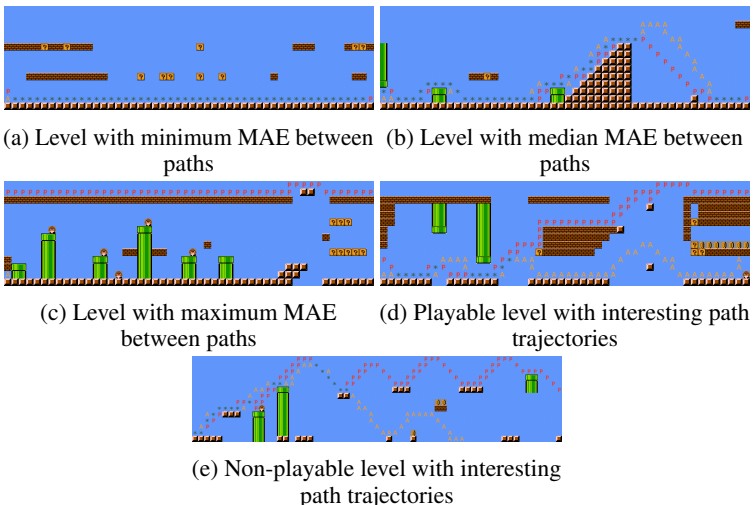

(a) Level with minimum MAE between paths

(b) Level with median MAE between paths

(c) Level with maximum MAE between paths

(d) Playable level with interesting path trajectories

(e) Non-playable level with interesting path trajectories

Figure 5: A* vs. MarioGPT generated paths. Levels with (**a**) minimum (0.02), (**a**) median (0.89) and (**a**) maximum (11.0) mean absolute error (MAE) between trajectory of actual A* agent (denoted as A), and model suggestion (denoted as P), as well as interesting hand-picked examples. Positions where both trajectories overlap are marked with *. Paths suggested by the model generally tend to have more airtime than the A* agent (**d**, **e**), likely due to game physics not being accounted for in the original path annotations of the training data.

## 4.3 Is MarioGPT memorizing?

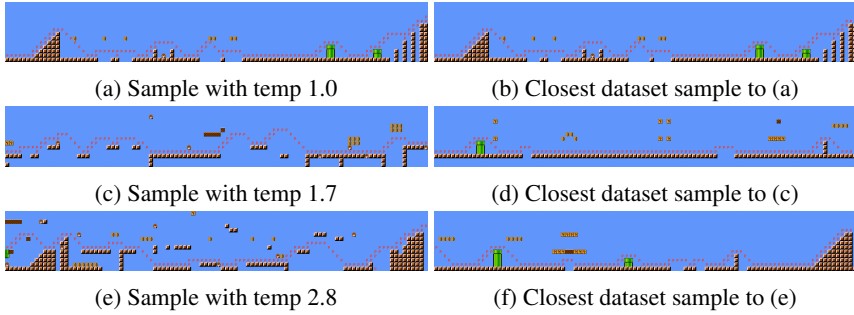

(a) Sample with temp 1.0

(b) Closest dataset sample to (a)

(c) Sample with temp 1.7

(d) Closest dataset sample to (c)

(e) Sample with temp 2.8

(f) Closest dataset sample to (e)

Figure 6: Generated levels vs closest in dataset. Temperature of 1.0 ends up spitting out almost exactly what is in the dataset, while increasing temperature improves sample diversity.

Memorization dynamics in LLMs remain an open problem when training transformer architectures [41, 5, 16]. While LLMs are incredibly powerful, they can sometimes overfit extremely and end up regurgitating training data. One popular way to alleviate this issue is to add some randomness in predictions in the form a tunable "temperature" parameter [17]. To evaluate whether MarioGPT is generating levels that are identical to the training set, we sample with different temperature parameters

and compare them the closest level in the training dataset. From Figure 6, we can see that increasing temperature results in samples that are more diverse, but lack quality. In our case, when generating levels we use a temperature of 2.4-2.7, as it can generate diverse samples while still retaining some quality. There are many possible improvements to explore in the future. One common way is to simply increase the richness of the dataset. The more samples the model has access to, the less likely it is to overfit. We could also improve MarioGPT's sampling abilities by introducing different search methods other than sampling with temperature, such as constrained beam search [6] and dataset augmented search [16], to increase diversity while preserving more quality.

## 4.4 Guided Level Generation via Prompting

Through simple prompting, we are able to guide MarioGPT towards controllable and diverse level generation. We empirically evaluate the prompting ability of MarioGPT by generating 1,000 samples with various combinations of prompts, and check how accurate the generated levels are to the prompt descriptions. The results suggest that MarioGPT can generate levels that match their given prompts most of the time (Table 3). MarioGPT is the most accurate with blocks and the least accurate with enemies. This is expected because there are fewer total tiles of enemies, while there are many more block tiles observed during training.

Table 3: Prompt vs actual description accuracy

| pipes | enemies | blocks | elevation |
|-------|---------|--------|-----------|
| 81%   | 68%     | 92%    | 76%       |

We visually evaluated the system, displaying selected prompt-conditioned generations in Figure 1. In addition, we evaluate the importance of the keywords in the prompt by comparing the distribution of the number of pipes between levels generated with random prompts without pipes-related commands (e.g. "some enemies, some blocks, high elevation") versus random prompts with pipes-related commands (e.g. "little pipes, some enemies, some blocks, high elevation"). The distribution without pipe prompts is scattered, while the ones with pipe prompts result in distributions with peaks, indicating that the keywords actually have an effect on the level generated (see Figure 11 in the Appendix).

MarioGPT is also able to generate levels from text descriptions that are not represented in the dataset. For instance, Figure 1e shows a successful approximation of the prompt, "many pipes, no enemies, many blocks", with a slight inaccuracy in that it has 1 less pipe (5 pipes is considered "many", while 4 are present). However, this is not always the case, as can be seen in Figure 1f, where the model, prompted by "many pipes, no enemies, some blocks", generates a level with the correct number of pipes and blocks but generates too many enemies. In future work, we hope to explore more ways to incorporate prompt importance, such as editing levels with tiles to create more samples or prompt tuning [18].

## 4.5 Generating Diverse Levels with Novelty Search

Through the combination of an LLM (Section 3.1) and novelty search (Section 3.2), we are able to continuously generate diverse levels in an open-ended fashion. Specifically, NS-MarioGPT is able to generate a collection of levels with a diverse set of predicted agent paths. We project the archive as a set of 2D embeddings in Figure 7 and darken the embedding points that are added later in the process. We can see that the levels are increasingly filling up empty spots in the embedding space. We also compare the distribution of levels generated by novelty search to levels generated by random prompts in Figure 8a. Visually, we can see that the levels generated by novelty search are more spread out in t-SNE space and the sampled ones, indicating that they are more diverse. Finally, we have also evaluated the playability of levels generated by novelty search, and find that the majority are solvable and non-playable levels are not clustered but rather scattered across t-SNE space. This indicates that there is no trade-off between path diversity and the ability to generate solvable levels. Figure 8b shows the corresponding results.

Figure 9 displays all the overlayed predicted paths (in a level grid) as more and more levels get added to the archive during novelty search. Similar as in Figure 7, we can see that over time, the space

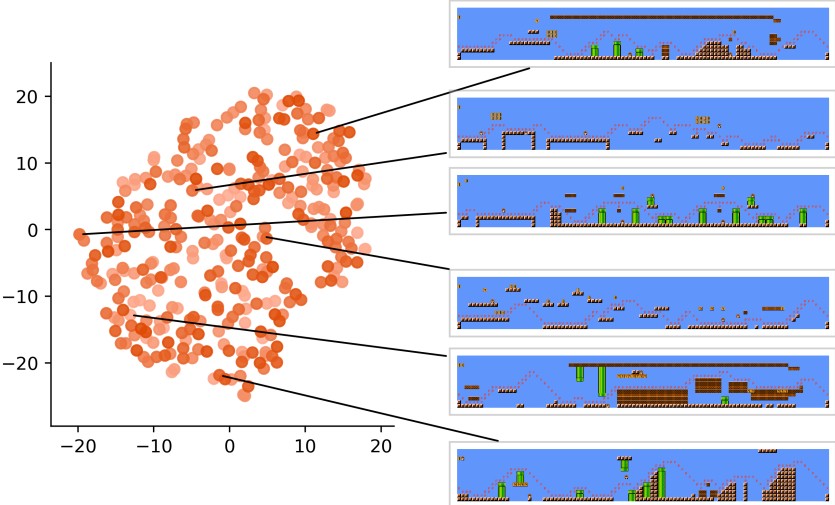

Figure 7: t-SNE of the levels in the archive. t-SNE embeddings are computed from the behavioral characteristic. Darker points indicate more recently added elements. Although novelty search is using the behavioral characteristics of the player paths, the levels also demonstrate visual novelty.

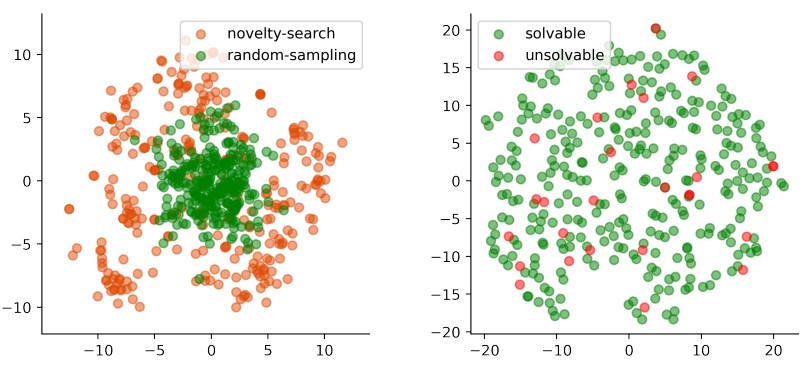

(a) Novelty search vs. random sampling      (b) Playable vs. non-playable levels

Figure 8: Comparing exploration for novelty search vs. random sampling and playable vs. non-playable levels. (**a**) t-SNE of both the embeddings of novelty-search levels and levels generated with random prompts. The visualization suggests that novelty search enables a much wider exploration of the space of levels. (**b**) Unsolvable levels are not clustered together but instead scattered across the t-SNE space. This distribution indicates that there is no correlation between the diversity of levels and their solvability.

of possible predicted agent paths gets filled, as increasingly diverse levels are mutated and added to the archive. As more levels are added to the archive, more and more of the tiles / empty space in the grid are being filled up, indicating that NS-MarioGPT is discovering a variety of levels that produce diverse paths. Concretely, we found that after 300 levels are added to the archive, around 78% of the possible coordinates are filled up. However, there are still many overlapping paths in the archive, meaning that similar paths are still being added to the archive. This is an issue that could be improved by using more related time series distance metrics that account for patterns in a path [11].

Levels with the highest and lowest novelty score from the archive are also shown in Figure 10. The level with the lowest novelty, shown in Figure 10a, has a path that is much more common in the archive, which can be seen by its almost identical look compared to the 2nd lowest in Figure 10c. The levels with higher novelty, Figure 10b and Figure 10d, have more unique patterns, but share a similar

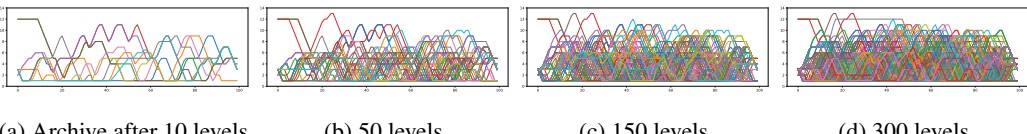

| (a) Archive after 10 levels | (b) 50 levels | (c) 150 levels | (d) 300 levels |

Figure 9: Generated path populations during novelty search. Each line is a path through a level. Over time, NS-MarioGPT fills more and more of the space of all possible paths.

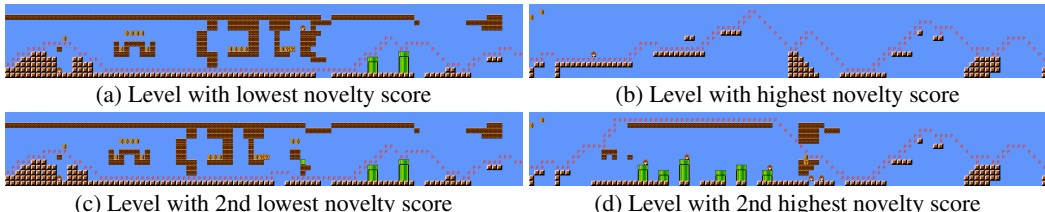

| (a) Level with lowest novelty score | (b) Level with highest novelty score |
| (c) Level with 2nd lowest novelty score | (d) Level with 2nd highest novelty score |

Figure 10: Comparison of most and least novel levels in the archive. The two least novel levels are very similar to each other, while the most novel levels have more distinct path patterns.

pattern towards the end. This indicates that one was created by mutating the other. We also found that the diversity starts to plateau after around 350-400 generations. However, this is very sensitive to the behavior characteristic (the smoothed predicted path of the level), so it may be different for other behavior characteristics.

While NS-MarioGPT is still able to discover many diverse levels through its simple mutation process, more complex functions could also be explored. For instance, crossover, a common mutation utilized in many genetic algorithms, would increase mutation diversity which can lead to more diverse levels.

## 5   Conclusion

Here we introduced MarioGPT, a fine-tuned GPT2 LLM that can not only generate diverse levels, but can guide its generation via a language prompt. This ability is useful in the field of Procedural Content Generation, where balancing controllable and diverse generation is a difficult task. We showed that MarioGPT is also able to (1) predict player interaction in generated levels, (2) generate diverse and **playable** environments, and (3) reduce the need for expensive external agent interactions (MarioGPT can generate playable levels approximately 88% of the time). Additionally, when combined with a diversity-driven algorithm like novelty search, MarioGPT can generate open-ended and functional content. While MarioGPT can generate diverse content, it is still limited. It can sometimes not follow prompt instruction and it is also subject to memorizing the dataset. We hope to improve these limitations by introducing richer data to the system.

One major benefit of re-using an existing LLM is that we can take advantage of all the research and improvements that have gone into these architectures. We plan to leverage the scalability of LLMs and train MarioGPT on bigger, more detailed annotated levels. Also, we are particularly excited about incorporating human feedback into the level generation process through reinforcement learning from human feedback (RLHF) [47]. The ability to fine-tune these models on human feedback allows users to continually tune their generated levels towards desired characteristics. Ultimately, we hope that MarioGPT opens the door to more controllable and diverse PCG systems.

## Acknowledgements

This project was supported by a DFF-Research Project1 grant (9131- 00042B) and a European Research Council (ERC) grant (GA no. 101045094, project "GROW-AI").

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

## 6 Appendix

### 6.1 Dataset Details

Table 4: Unique Mario tiles

| Tile Type | Symbol | Visualization |
|-----------|--------|---------------|
| Empty | - | |
| Unbreakable | X | |
| Breakable | S | |
| Question Block | ? / Q | |
| Coin | o | |
| Enemy | E | |
| Left pipe top | < | |
| Right pipe top | > | |
| Left pipe lower | [ | |
| Right pipe lower | ] | |
| Cannon Top | B | |
| Cannon Body | b | |
| Path | x | |

### 6.2 Constructing Prompts

Prompts are represented as combinations of specific features (e.g. pipes, enemies, blocks, elevation) alongside quantitative keywords:

- { *no, little, some, many, [0-1000]*} *pipes*
- { *no, little, some, many, [0-1000]* } *enemies*
- { *little, some, many, [0-1000]*} *blocks*
- { *low, high*} *elevation*

As an example, "*no pipes, many enemies, low elevation*" or "*many pipes, many enemies, many blocks*" are both possible prompts. The keywords "no", "little", "some", "many" are calculated from quantiles of the corresponding count within a 50 column window (Table 5). The "low" and "high" elevation are determined from the height of the highest unbreakable blocks in a segment of the level.

Table 5: Prompt Quantiles and corresponding counts within a 50 column window

| tile | no | little | some | many |
|------|-----|--------|------|------|
| pipes | 0 | 1 | 2 | 5 |
| enemies | 0 | 1 | 3 | 7 |
| blocks | 0 | 50 | 75 | 176 |

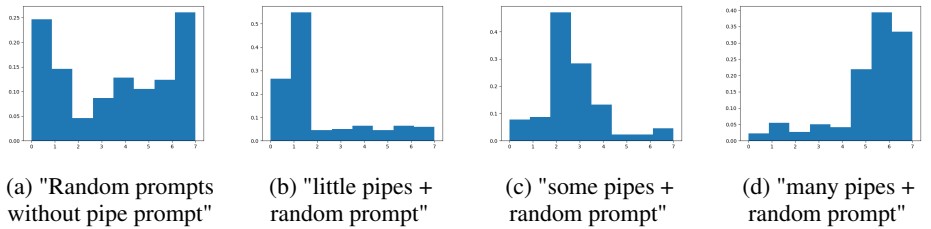

(a) "Random prompts without pipe prompt"

(b) "little pipes + random prompt"

(c) "some pipes + random prompt"

(d) "many pipes + random prompt"

Figure 11: Effect of prompt conditioning. Comparison of the distribution of the number of pipes between levels generated with random prompts without pipes-related commands (e.g. "some enemies, some blocks, high elevation") versus random prompts with pipes-related commands (e.g. "little pipes, some enemies, some blocks, high elevation"). The distribution without pipe prompts is scattered, while the ones with pipe prompts result in distributions with peaks.

