# OpenReview forum: "MarioGPT: Open-Ended Text2Level Generation through Large Language Models"
_NeurIPS.cc/2023/Conference — NeurIPS 2023 poster_

### Official Review · Reviewer_an9Y · 2023-07-04

**Soundness:** 3 good
**Presentation:** 3 good
**Contribution:** 2 fair
**Rating:** 6
**Confidence:** 4

**Summary:**

This paper proposes MarioGPT with novelty search, a method that can generate new Super Mario Bros levels. MarioGPT is finetuned from GPT-2 to generate levels. The novelty search is used to get novel levels by randomly selecting a generated level, mutating it, and then filtering it based on a novelty criterion: the mean distance between the mutated level and the K closest elements from the existing levels (K-means). Experimental results show that MarioGPT has a better level of reconstruction accuracy than LSTM, and 88.33% of the generated levels can be played by Robin Baumgarten’s A* agent.

**Strengths:**

1. The idea of generating Super Mario Bros levels is interesting and might be helpful for both the reinforcement learning community and the game community.

2. The result shows that the finetuned GPT has the ability to generate decent Super Mario Bros levels, which is insightful.


**Weaknesses:**

1. The evaluation criteria do not seem to be appropriate. For example, "playable by an A* agent" is not a convincing indicator that can measure the quality of generated levels. Actually, "playable by an A* agent" primarily measures the simplicity of the generated levels rather than the quality of the levels. Moreover, in section 4.4, "Guided Level Generation via Prompting", the "empirical evaluation" is not objective, and lacks baseline (like LSTM) comparisons. Since this paper claims that the generated levels are text-controllable, it would be better to provide a more explicit demonstration of controllability.

2. The baseline (LSTM) in section 4.1 is weak. Table 1 shows that the pre-trained GPT-2 is very important in the generation process. Then, how does a GPT-2 without finetune perform? How does GPT-3/3.5/4 perform? Also, I doubt the conclusion that LSTM is not promptable since it can be integrated with a text encoder.

3. Although the idea is interesting, the generated levels are simple. 88.33% of the generated levels can be played by an A* agent, and only horizontal movements are included (Mario cannot crawl into the pipes and there is no hidden brick, making the generated levels even simpler than traditional Super Mario Level 1-1). This greatly reduces the significance of this work, both in the fields of artificial intelligence and gaming industry.


**Questions:**

1. How to control detailed level settings like the initial moving direction of the enemy?

2. Can MarioGPT be tuned on Super Mario Makers which consists of thousands of novel levels contributed by the gaming community?

3. Why do from-scratch-MarioGPT and adapter-MarioGPT perform worse than LSTM?

4. Mario GPT generates trajectories as well as game contexts. What about generating trajectories first, then selecting a trajectory and keeping it unchanged, and then generating, mutating, and filtering the game contexts? Will this improve the “playable rate” of the generated games?


**Limitations:**

See above questions.

---

> ### Author Rebuttal · Authors · 2023-08-10
>
> We thank the reviewer for the valuable feedback!
>
> **Addressing the qs from the reviewer**
>
> > How to control detailed level settings like the initial moving direction of the enemy?
>
> Because of the limitations of the current dataset and its labels, we do not know the initial direction of the enemy. However, our system can scale with more diverse prompts and we plan to explore that front in future work, touching upon more semantic and visual descriptions of levels.
>
> > Can MarioGPT be tuned on Super Mario Makers which consists of thousands of novel levels contributed by the gaming community?
>
> GPT2 has been shown to be a very scalable architecture and can digest large amounts of diverse data. Given this, we believe that our architecture can scale with increasingly diverse levels (such as those generated by Super Mario Maker). We hope to explore this more in future work.
>
> > Why do from-scratch-MarioGPT and adapter-MarioGPT perform worse than LSTM?
>
> We hypothesize that the parameter space of MarioGPT-from-scratch is quite large and makes it harder to optimize. That being said, we simply utilized the same hyperparameters presented in the DistilGPT2 work: https://tinyurl.com/distillcode, so there may be some improvements to be made with a more optimized hyperparameter search and longer training. Tangentially, this highlights one benefit of finetuning, as there was little to no hyper parameter tuning.
>
> > What about generating trajectories first...?
>
> I’s definitely possible to mask out all other parts of the level and keep the path tiles, allowing MarioGPT to inpaint the level around the desired path. To use this, we would probably need to train a separate network to just generate “playable” paths, which could perform the same (or worse) than the current system. That being said, this inpainting ability can still be very useful for path customization; we plan to look more into this in future work.
>
> **Addressing the weaknesses pointed out by the reviewer**
>
> > ..."playable by an A* agent" is not a convincing indicator that can measure the quality of generated levels...
>
> For procedural content generation of Super Mario Levels (and many other tile-based games), A* is in fact the standard when it comes to measuring playability (see references [1-5])
>
> The reason for choosing Robin Baumgarten’s A* agent for measuring playability comes from its performance on the 2009 Mario AI competition, where it beat handcrafted controllers and even simple evolved neural networks on getting the furthest in an infinite-level setting, as well as solving a corpus of levels [6]. To summarize, A* is able to solve difficult Mario levels beyond human performance (we refer the interested reader to a video of the A* agent in question: https://tinyurl.com/astaragent).
>
> [1] Volz, Vanessa, et al. "Evolving mario levels in the latent space of a deep convolutional generative adversarial network." Proceedings of the genetic and evolutionary computation conference. 2018.
>
> [2] Awiszus, Maren, Frederik Schubert, and Bodo Rosenhahn. "TOAD-GAN: Coherent style level generation from a single example."
>
> [3] Schrum, Jacob, Vanessa Volz, and Sebastian Risi. "Cppn2gan: Combining compositional pattern producing networks and gans for large-scale pattern generation."
>
> [4] Volz, Vanessa, et al. "Tools for Landscape Analysis of Optimisation Problems in Procedural Content Generation for Games."
>
> [5] Edwards, Maria, Ming Jiang, and Julian Togelius. "Search-based exploration and diagnosis of TOAD-GAN."
>
> [6] Togelius, Julian, Sergey Karakovskiy, and Robin Baumgarten. "The 2009 mario ai competition."
>
> > ...It would be better to provide a more explicit demonstration of controllability
>
> We compared levels that are prompted without a “pipes” description to those that specify either “little”, “some”, “many” pipes (see Figure 1 of rebuttal pdf). We found that without specifying pipes, the distribution of the number of pipes that MarioGPT generates is scattered, while specifying keyword descriptions results in distributions that match. This shows that prompt-conditioning has a significant effect on level generation. We saw similar behavior for the other tile categories as well.
>
> > ...how does a GPT-2 without finetune and how does GPT-3/3.5/4 perform?
>
> GPT2 and GPT3.5 cannot generate legitimate mario levels when prompted, even when giving few shot examples and tile descriptions. This is because GPT2 turns out to be not powerful enough for complex in-context learning, and GPT3.5 seems to hallucinate and output random tiles that do not correspond to any prompt the user has given.
>
> The reviewer pointed out a great point about the LSTM: The LSTM is not quite promptable in the same way MarioGPT is, (there’s no real attention mechanism), but we have included a simple way to prompt (by appending the prompt embeddings to input). Even with this, the LSTM ends up only generating ~31% playable levels.
>
> > Although the idea is interesting, the generated levels are simple..
>
> With regards to the simplicity of levels (reviewer pointed out that there are no pipe interactions + hidden bricks): Some of the generated levels are still quite difficult for the A* agent, which only clears about 87% of the original Super Mario Bros and Super Mario Bros 2. The generated levels are also quite complex for human players, some requiring finely timed jumps and sharp increases / decreases in elevation.
>
> The generated levels, especially when combined with Novelty Search, are in fact often quite complex (see Figure 8). Hidden bricks and similar elements could be added easily by adding denoting characters to the training levels, upon which the model would learn an appropriate token, which can then also be implemented in the simulator parsing the MarioGPT output. In general, as long as complex mechanisms can be converted to tokens and reflected in simulation, MarioGPT should be capable of generating them.

---

> > ### Author Response · Authors · 2023-08-17
> >
> > We hope we were able to address the concerns of Reviewer an9Y? We believe we addressed the main criticism of (1) using A* as an evaluation metric, and (2) evaluating against other baselines.
> >
> > If there are any other questions or issues, please let us know.

---

> ### Comment · Reviewer_an9Y · 2023-08-18
> **Thanks for the rebuttal**
>
> Thanks for the rebuttal and response to my questions. Most of my concerns have been addressed, especially the comparison baselines. I will raise my score to 6.

---

### Official Review · Reviewer_aEvk · 2023-07-05

**Soundness:** 3 good
**Presentation:** 4 excellent
**Contribution:** 3 good
**Rating:** 6
**Confidence:** 4

**Summary:**

This paper proposes a novel idea of using LLMs to perform Procedural Content Generation. It addresses several challenges including the diversity of the generated environments, the existence of a feasible solution (playability) and generating with language guidance. It proposes two key algorithms.

1. MarioGPT - Fine-tuning on DistilGPT2 so that it can predict the next slice of the game. It also allows prompting - the text conditioning is done by using BART to encode text prompts then feeding the average hidden states into the cross attention.

2. Novelty search for open-endedness - Using MarioGPT as a mutation operator in the novelty search evolutionary loop, prompting the MarioGPT with random prompts and generate candidates, and retain the candidates that pass the novelty criteria to add to the archive. Then do masked token prediction (impainting) to ensure that the level generated has a feasible path solution.

This paper conducts experiments that demonstrate (1) the high token prediction accuracy of MarioGPT, (2) good playability - i.e. most generated levels have feasible path solutions, (3) using a suitable temperature to control the trade-off between diversity and quality, (4) the accuracy in following the prompt and (5) the diversity of generated levels (in terms of the predicted path) achieved by novelty search.

**Strengths:**

1. This paper proposes a novel framework for text-guided PCG: having a text-conditioning, generative model that predicts new levels, then having a novelty search algorithm reject the levels that are too similar to existing ones.

1. This paper is written clearly. The methods are well motivated. The experiments are well-designed with detailed analysis and discussion.

**Weaknesses:**

1. Experiments are lacking in measuring “controllability”. The prompts are limited to a small set. The capability of following instructions on a more semantic level are not measured, such as the scene layout. and the spatial, functional, and semantic relationships between objects. (e.g. how tall the pipes are, “there is an enemy to the right of a pipe”).

2. The technique of first auto-regressively generating a scene, then using “inpainting” to ensure a feasible solution seems to be pretty tied to this specific type of games where the scene moves linearly in 1 dimension (i.e. the character can only move forward, not backward), and hard to generalize to other PGC scenarios. For example, in order to generalize to the cases where the character can move both forward and backward, non-trivial changes are needed.

**Questions:**

1. In section 4.4 "Guided Level Generation via Prompting", under what sampling temperature are these results measured? I am curious how / whether this randomness affects how accurately the model follows the prompts as well. The authors mentioned that randomness helps with diversity - does this sacrifice accuracy of following prompts?

2. On a relevant note, in section 4.3 the authors mentioned that the temperature controls the tradeoff between diversity and quality. Could you elaborate what the definition of "quality" is?

3. In section 4.5, at what point does the “diversity” start to plateau? Are there baseline studies regarding the diversity of generated paths?

4. In section 4.2, are there quantitative baselines regarding the playability of generated environments? Could you share intuition regarding the trade-offs between playability versus diversity of the generated environments?

**Limitations:**

As mentioned above, (1) the measurement of capability to understand the instructions semantically and (2) the generalizability of this approach beyond this type of games remains to be further studied.

---

> ### Author Rebuttal · Authors · 2023-08-09
>
> Thanks for the great feedback!
>
> **Addressing the questions from the reviewer**
>
> > In section 4.4 "Guided Level Generation via Prompting", under what sampling temperature are these results measured? ... The authors mentioned that randomness helps with diversity - does this sacrifice accuracy of following prompts?
> On a relevant note, in section 4.3 the authors mentioned that the temperature controls the tradeoff between diversity and quality. Could you elaborate what the definition of "quality" is?
>
> These are some great points brought up by the reviewer. Large Language Models, especially those that are trained on smaller datasets, like the set of levels we used in our work, suffer from overfitting and even “memorization”. This can be seen in Figure 7, where a temperature of 1.0 effectively results in the model spitting out a level (or slice of a level) that exists in the dataset. To alleviate this, temperature can be used as a toggle-able hyperparameter to control the degree of randomness or uncertainty in predictions. This effectively changes the skewed distribution of token predictions (such as a very large logit value for a single token), to a more smoothed distribution. This has an important tradeoff – more randomness introduces more diversity in tokens at the cost of “tile accuracy”, which means that increasing temperature can reduce the impact of the prompt and the effect of previous contexts. This is what we mean by “quality diversity” tradeoff, where "quality" here indicates how "natural" the level is, or how similar are its characteristics to the original levels. In section 4.4, we utilize a common temperature value of 1.4. We will make sure to clarify this in the paper.
>
> > In section 4.5, at what point does the “diversity” start to plateau?
>
> We found that the diversity starts to plateau after around 350-400 generations. However, this is very sensitive to the behavior characteristic (the smoothed predicted path of the level), so it may be different for other behavior characteristics.
>
> > Are there baseline studies regarding the diversity of generated paths?
>
> With regards to baseline studies of diversity of generated paths, we include a t-SNE plot of levels generated by novelty search along with levels that are generated from randomly sampled prompts (Figure 2 in rebuttal pdf). The spread of levels generated by novelty search is much larger than the levels generated by sampling, indicating that novelty search results in a more diverse set of generated levels.
>
> > In section 4.2, are there quantitative baselines regarding the playability of generated environments? Could you share intuition regarding the trade-offs between playability versus diversity of the generated environments?
>
> We found that the best baseline, the LSTM, achieves around ~31% solvable levels, suggesting that MarioGPT is much better at generating solvable levels (achieves ~88%). With regards to the trade-off between diversity of generated levels (from novelty search) and playability, we plotted TSNE embeddings (in Figure 3 of the rebuttal pdf) and labeled “green” for solvable and “red” for unsolvable. We can see that the unsolvable levels are spread across the space without a discernible pattern, indicating that there isn’t really a relation between diversity of levels and playability. This is expected, because our mutations consist of just resampling portions of the level (with a random prompt) and stitching it together with the inpainting model. These models both learn how to generate valid and playable levels, so the mutated portions follow the same.
>
> **Addressing the weaknesses pointed out by the reviewer**
> > Experiments are lacking in measuring “controllability”.  ... The capability of following instructions on a more semantic level are not measured, such as the scene layout. and the spatial, functional, and semantic relationships between objects. (e.g. how tall the pipes are, “there is an enemy to the right of a pipe”).
>
> This is a great point! For evaluating controllability, we included a new experiment (detailed in Figure 1 of the rebuttal pdf), where we compare random prompts excluding pipe descriptions, to those with pipe descriptions. The distribution of number of pipes for levels where pipes are excluded in the prompts is very spread out, while the other ones have peaks corresponding to the description. This indicates that prompting does have an effect, and it's possible to control the generation.
>
> Our prompts are currently simple (only dealing with counts of objects), but we still see that they can guide the model towards generating a diverse set of levels. Additionally, given how language models are able to scale well with richer data, we believe that our system can scale to many more detailed prompts like the ones mentioned by the reviewer.
> For future work, we not only want to explore more of these semantic descriptions mentioned by the reviewer, but also visual descriptions of the level. This is an exciting direction and we hope to explore more in the future!
>
> > The technique of first auto-regressively generating a scene, then using “inpainting” to ensure a feasible solution seems to be pretty tied to this specific type of games where the scene moves linearly in 1 dimension (i.e. the character can only move forward, not backward), and hard to generalize to other PGC scenarios...
>
> In regards to the reviewer’s concern about the generality of the inpainting method: Because both the tile prediction model and the inpainting model work on character level predictions, they are actually robust to any orientation. For instance, for a game like Sokoban, where movements can be in any direction, the level is still composed of tiles that can be passed into the LLM one by one. Figuring out the "borders" after resampling a portion of the level and inpainting the surrounding tiles also does not require much change in the system.

---

> > ### Comment · Reviewer_aEvk · 2023-08-22
> > **Change contribution from Fair to Good**
> >
> > Thank you for the rebuttal and detailed, helpful responses to my questions. I am keeping the rating, and raising the contribution from 2.Fair to 3.Good, because the added LSTM baseline highlighted your new models' capabilities of generating playable levels. I look forward to your future work on controlling the generated environment with more sophisticated prompts. Thank you again! :)

---

### Official Review · Reviewer_wPh9 · 2023-07-07

**Soundness:** 3 good
**Presentation:** 3 good
**Contribution:** 3 good
**Rating:** 7
**Confidence:** 3

**Summary:**

In this work the authors fine tune a DistilGPT2 model to produce diverse, largely playable Mario levels. They incorporate a novelty search to favor diversity in levels, and also explore conditioning level generation on natural language user prompts. Their novelty search proceeds as follows: Given an initial set of levels, a level is sampled from the set and randomly mutated. The mutation involves deleting a random contiguous segment of the level and sampling a new one in its place from their model, then using a fine tuned BERT model to do inpainting on the border regions to smooth the transition between this and the rest of the level. The mutated level is accepted if it is sufficiently different from the other levels, using a distance metric comparing the player path through the level.

**Strengths:**

- The use of LLMs for game level generation certainly makes sense, and I think it's valuable to publish work in this area for the NeurIPS community to see and build on. The technical approach taken here is original, to my knowledge.
- The combination of genetic algorithms with LLMs is interesting and novel, and I found the use of BERT for inpainting to smooth out the transition between the mutated segment and the rest of the level to be a particular clever approach.
- The paper also does a great job of visualizing the levels produced by the model which is very helpful in getting a feel for the sorts of outputs that their model produces. The playability and temperature analyses were quite interesting as well.

**Weaknesses:**

- Section 4.5: It's hard to tell that the novelty search is actually helping, without a comparison to an ablated version that doesn't do the novelty search. In particular it's hard to tell if the darker points in Figure 9 seem to be filling in the space in between because of the novelty search, or if it's just that there are more sampled points so they happen to fill in some of the space in between the earlier points. Likewise with the rightmost graphs in figure 7 looking denser than the leftmost ones – taking more samples will always add some density, so in order to really conclude something you need a comparison to an ablation. Showing something like 300 levels from the novelty search plotted on the same tSNE as 300 levels from a version without novelty search (eg as a different color) would make it much clearer that there's actually an improvement in diversity (and likewise some analogous experiment for the other figure).
* In a similar vein, in Table 3 it's hard to know how much the prompt is helping without seeing the scores without prompting (for an idea of what scores a fairly random approach would achieve); though I think is a less major issue.
* After seeing the LSTM from prior work used in the Table 1 comparison, I was surprised that LSTM comparisons weren't given for any of the future evaluations, which would be helpful for understanding how much better this LLM setup is than prior work. For example, we learn that 88% of levels are solvable, but how does this compare to the LSTM approach? Likewise in the diversity analysis, a naive LSTM baseline would be interesting (but no need for all the novelty search machinery). This would just help me get a sense of where this work stands compared to prior work on these metrics that are being measured. That said, I wouldn't reject this paper on that basis – 88% playable is a great statistic regardless of what you compare it to, and I don't actually expect that the LSTM would make particularly good levels, but having the full comparison would strengthen the evaluation.
- As a last, minor point, It's surprising (in a way that makes me feel that an explanation is needed) that such poor performance is obtained by from-scratch-MarioGPT on the tile prediction task given it was trained on 200,000 datapoints. The "tile accuracy" is presumably a measure of how well the system does next-tile prediction (analogous to next-token prediction in language modeling, and presumably this is the objective that all these models are trained on). I'd imagine something as simple as "always predict an air tile" since most levels look like they're more than half air, or "predict the same tile that you saw to the left" would do quite well, and the inability to pick up on even simple trends like this after 200,000 datapoints is surprising to me. Perhaps there is additional complexity I'm missing?

Overall, this is an interesting paper that I end up borderline reject on because of the evaluation points mentioned above, but I don't have a huge amount of prior experience in this area or what typical evaluations are for PCG, so I am open to revising through discussion.

**Questions:**

* Relating to the weakness point above, in Table 1 I'm a little surprised that everything except MarioGPT does so poorly (LSTM does okay at least though). It'd be helpful to have a little intuition for why this is so hard?
- The concurrent work cited by this paper (Todd et al 2023) finds that pretraining (vs from-scratch) GPT2 has little to no impact on a range of metrics. That is of course a somewhat different task and setup, but I'm curious if you have thoughts on why you seem to find the opposite result in this domain?
- What embedding is used for the tSNE? Is it something like the last layer of MarioGPT?
- Do you guarantee that each tile is parsed as a separate token? Without this it seems like the network could potentially get confused about reasoning about lengths/distances, like if one token represented two blocks while another represented just one. Seems like it works either way, but I'm curious if this is something you're doing or might look at.
- The language conditioning is cool! Of course, a natural extension of this would be moving towards level generation from more unstructured natural language. For example, flexibility with synonyms to "many" or more complex things like "a series of narrow pillars with an enemy on top of each one". That sounds like enough for a separate paper but I'm personally curious if you've done any preliminary work in that direction or what you think of it. To be clear, I don't think this is needed in this submission I just think that the work is cool and am curious!

**Limitations:**

I think that this paper does an adequate job of presenting its limitations.

---

> ### Author Rebuttal · Authors · 2023-08-09
>
> First off, we thank the reviewer for the great and thoughtful feedback!
>
> **To answer the reviewer’s open questions:**
>
>  > 1. "in Table 1 I'm a little surprised that everything except MarioGPT does so poorly (LSTM does okay at least though). It'd be helpful to have a little intuition for why this is so hard?”
>
> We hypothesize that the parameter space of MarioGPT-from-scratch is quite large and makes it harder to optimize. That being said, we simply utilized the same hyperparameters presented in the DistilGPT2 work: https://github.com/huggingface/transformers/blob/main/examples/research_projects/distillation/train.py, so there may be some improvements to be made with a more optimized hyperparameter search. We also hypothesize that extending the training process may also help. The LSTM seems to have an easier time, possibly because of the smaller parameter space.
>
> > 2. “The concurrent work cited by this paper (Todd et al 2023) finds that pretraining (vs from-scratch) GPT2 has little to no impact on a range of metrics. That is of course a somewhat different task and setup, but I'm curious if you have thoughts on why you seem to find the opposite result in this domain?”
>
> We hypothesize that this is because of some notable differences to our work and the approach in the work by Todd et al (2023). In Todd et al. the text prompts and the generation of the level are done together (as opposed to using a frozen text encoder like our approach), which may have had an effect on performance. In addition, generating Sokoban levels could pose other challenges compared generating Super Mario Bros levels, which we will also explore in more detail in future work.
>
> > 3. “What embedding is used for the tSNE? Is it something like the last layer of MarioGPT?”
>
> The embedding space is actually the smoothed moving average of the predicted path, which is essentially a time series (can be seen in Figure 4). We will make this clearer in the paper.
>
> > 4. “Do you guarantee that each tile is parsed as a separate token?”
>
> This is a great point! We verified this and each tile is indeed parsed as a separate token. We will make sure to add this detail in the paper.
>
> > 5. ...a natural extension of this would be moving towards level generation from more unstructured natural language. For example, flexibility with synonyms to "many" or more complex things like "a series of narrow pillars with an enemy on top of each one".
>
> We did some preliminary experiments that show it is possible to use synonyms for words. For example, changing “many” to “a lot” or “a ton”, produces similar results. For more complex ones like the one mentioned by the reviewer ("a series of narrow pillars with an enemy on top of each one”), we wouldn’t expect the currently fined-tuned MarioGPT to work well. However, the approach should be able to generalize to such more complex prompts by keeping the MarioGPT architecture the same but adding diversity the annotated training set. Currently there’s no easy way to add “visual” descriptions of levels yet, but there are many exciting approaches that we hope to explore in future work, like utilizing visual language understanding models.
>
> **Addressing the weaknesses pointed out by the reviewer:**
>
> > “Section 4.5: It's hard to tell that the novelty search is actually helping, without a comparison to an ablated version that doesn't do the novelty search. ”
>
> This was a very valuable comment and as suggested, we now created an archive by generating levels using randomly sampled prompts (as shown in green in Figure 2; see PDF attached to the rebuttal summary at the top). We combined the novelty search archive and the sampled archive and ran t-SNE on the combined archive. Visually, we can see that random sampling actually results in a decently diverse set of levels (indicated by the spread of points in t-SNE space). However, our novelty search approach clearly results in a much more diverse set of levels, as can be seen by the spread of points shown in orange.
>
> > “... in Table 3 it's hard to know how much the prompt is helping without seeing the scores without prompting”
>
> This was a very valid point and we have now investigated the effect of prompt conditioning in more detail, which is shown in Figure 1 in the PDF attached to the main rebuttal summary. In this figure, we compared the distribution of the number of pipes between levels generated with random prompts without pipes-related commands to random prompts with pipes-related commands The results demonstrate that the prompt-conditioning has a significant effect on the tile distribution. We see similar results for the other tile types.
>
> > “... we learn that 88% of levels are solvable, but how does this compare to the LSTM approach?”
>
> Running the same evaluation setup as MarioGPT (over 250 randomly sampled prompts), we found that the LSTM generates playable levels ~31% of the time.
>
> > “As a last, minor point, It's surprising (in a way that makes me feel that an explanation is needed) that such poor performance is obtained by from-scratch-MarioGPT on the tile prediction task given it was trained on 200,000 datapoints. ”
>
> To clarify, tile accuracy here refers to non-air tiles. As pointed out by the reviewer, a majority of tiles are air tiles, and each model ends up heavily predicting air tiles, so we focus primarily on their ability to predict non-air tiles. However, there may be better metrics to measure time predictions here, namely those that can deal with in-balanced class datasets.
>
> As mentioned in the response to the 1st question above, MarioGPT-from-scratch performance can probably be improved with more time and optimized hyperparameter search. However, as a tangential point, this shows a major benefit of fine-tuning pretrained models, which seem to require much less effort in regards to hyperparameters. We will clarify these points in the in the paper.

---

> > ### Comment · Reviewer_wPh9 · 2023-08-16
> > **Changed Rating**
> >
> > Thank you for the extensive response and additional experiments/visualizations, I really appreciate this rebuttal. The addition of the first and third figures in the PDF that the authors attached make me much more convinced of the impact of the text conditioning and the novelty search. The additional baseline of LSTMs on the solvability objective is also helpful.
> >
> > My primary concerns in my initial review were around the soundness/thoroughness of the evaluation (baselines and ablations) and I feel the authors have dramatically improved that aspect of the paper, in addition to addressing my other thoughts. I believe that this would be a valuable paper for a NeurIPS audience, and have adjusted my rating to recommend Accept.

---

### Official Review · Reviewer_6gGD · 2023-07-07

**Soundness:** 3 good
**Presentation:** 4 excellent
**Contribution:** 3 good
**Rating:** 7
**Confidence:** 3

**Summary:**

This submission proposes MarioGPT, which aims to generate diverse-and-playable Mario levels with LLM (GPT-2) through language prompts. The input and output of MarioGPT are level representations, not natural language. Human prompts are involved by incorporating the cross-attention layer into the LLM. A Novelty Search algorithm is further proposed to increase the diversity of generated levels.

**Strengths:**

+ Leveraging LLM into Procedural Content Generation (PCG) is a natural idea. This submission successfully accomplishes this, and the proposed MarioGPT could be really usable.
+ This submission conducts a comprehensive study of various aspects of the proposed method, including the tile/path prediction accuracy, the playability of generated levels, whether MarioGPT is memorizing the training dataset, guidability by prompts. The overall experimental parts are convincing. The authors also analyze some current limitations and provide feasible solutions to them.
+ The proposed Novelty Search algorithm successfully improves the diversity of agent paths.

**Weaknesses:**

- Overall, I think Procedural Content Generation (PCG) with LLM is 'quite' an obvious idea, making me doubt this submission's 'technical' contribution to the community. I did not see any other obvious weaknesses.

**Questions:**

1. In the field of PCG, is there any game other than Super Mario? What is the generalization ability of the proposed method to other games (if there is any)?

**Limitations:**

Same as in the Questions part.

---

> ### Author Rebuttal · Authors · 2023-08-09
>
> We thank the reviewer for their thoughtful and useful comments!
> In terms of PCG applied to other games, while Super Mario Bros. is often used as a benchmark, there are certainly a variety of other games our approach could be applied to in the future. Another common PCG-benchmark are dungeon-like games such as the classic video game “The Legend of Zelda” (e.g. Khalifa et al. [1]). Our approach should generalize to such dungeon-like games as well because MarioGPT processes levels by flattening tile-based levels into long vectors, which is agnostic to content that can be represented as a string. Additionally, concurrently to our work, Todd et al. [2] showed that LLMs can also be used to generate levels for other games such as Sokoban, which indicates the generality of the approach.
>
> Future work could also focus on other types of content, usually represented as graphs or images (e.g. Doom levels [3] or StarCraft maps [4]). However, these non-string representation would require more substantial changes to the MarioGPT architecture.
>
> [1] Khalifa, A., Perez-Liebana, D., Lucas, S. M., & Togelius, J. (2016, July). General video game level generation. In Proceedings of the Genetic and Evolutionary Computation Conference 2016 (pp. 253-259).
>
> [2] Todd, G., Earle, S., Nasir, M. U., Green, M. C., & Togelius, J. (2023, April). Level Generation Through Large Language Models. In Proceedings of the 18th International Conference on the Foundations of Digital Games (pp. 1-8).
> [3] E. Giacomello, P. L. Lanzi, and D. Loiacono, “DOOM Level Generation using Generative Adversarial Networks.” arXiv, Apr. 24, 2018. [Online]. Available: http://arxiv.org/abs/1804.09154
> [4] Togelius, J., Preuss, M., Beume, N., Wessing, S., Hagelbäck, J., & Yannakakis, G. N. (2010). Multiobjective Exploration of the StarCraft Map Space. 2010 IEEE Symposium on Computational Intelligence and Games (CIG), 265–272. https://doi.org/10.1109/ITW.2010.5593346

---

> > ### Comment · Reviewer_6gGD · 2023-08-20
> > **Official Comments by Reviewer 6gGD**
> >
> > I appreciate the rebuttal, and the paper has addressed my questions and concerns. I will keep my rating 7 and hope the final version can contribute to the development of the field.

---

### Author Rebuttal · Authors · 2023-08-09

We would like to thank all the reviewers for their insightful comments, which helped us to significantly improve our paper. The most important new additions are: (1) a comparison to an ablated version without novelty search, which shows that novelty search is indeed crucial in allowing the discovery of a larger space of levels than random sampling; (2) we now also compare our approach to a prompt-able LSTM-baseline in terms of level playability and find that an LSTM-based approach generates playable levels around 31% of the time, which is significantly lower than the 88% for MarioGPT; (3) we clarify that A* is the current standard to evaluate procedurally generated levels and that solvable by A* does not adequate to levels being simple; (4) we added a controllability test to examine how much the text influences the generated levels, which shows that MarioGPT is indeed steerable through text prompts; and (5) We evaluated the playability of levels generated by novelty search, showing that the majority are solvable and there is no discernible tradeoff between path diversity and the ability to generate solvable levels. The new figures related to these additions can be found in the attached PDF.

---

### Decision · Program_Chairs · 2023-09-21

**Decision:**

Accept (poster)

**Comment:**

This paper proposed an LLM based approach for Procedural Content Generation, specifically for Super Mario Bros levels.

There has been concerns around the lack of technical contribution, the applicability to other content generation and the evaluation against baselines, while the authors have done a good job in convincing the reviewers during the rebuttal stages.

Based on the interesting application and also the potential interests from a broader community, I would recommend accepting this paper, while also encouraging the authors to incorporate the new figures/results obtained during the rebuttal stage to the camera ready version.